# Nitric Oxide Antagonism to Anti-Glioblastoma Photodynamic Therapy: Mitigation by Inhibitors of Nitric Oxide Generation

**DOI:** 10.3390/cancers11020231

**Published:** 2019-02-15

**Authors:** Jonathan M. Fahey, Albert W. Girotti

**Affiliations:** Department of Biochemistry, Medical College of Wisconsin, Milwaukee, WI 53226, USA; jonfahey5@gmail.com

**Keywords:** nitric oxide (NO), inducible nitric oxide synthase (iNOS), photodynamic therapy (PDT), glioblastoma PDT, NO-mediated PDT resistance

## Abstract

Many studies have shown that low flux nitric oxide (NO) produced by inducible NO synthase (iNOS/NOS2) in various tumors, including glioblastomas, can promote angiogenesis, cell proliferation, and migration/invasion. Minimally invasive, site-specific photodynamic therapy (PDT) is a highly promising anti-glioblastoma modality. Recent research in the authors’ laboratory has revealed that iNOS-derived NO in glioblastoma cells elicits resistance to 5-aminolevulinic acid (ALA)-based PDT, and moreover endows PDT-surviving cells with greater proliferation and migration/invasion aggressiveness. In this contribution, we discuss iNOS/NO antagonism to glioblastoma PDT and how this can be overcome by judicious use of pharmacologic inhibitors of iNOS activity or transcription.

## 1. Introduction

Malignant gliomas are the most prevalent primary brain tumors, and among these malignancies, glioblastoma multiforme (GBM), a grade-4 astrocytoma, is the most aggressive and resistant to a variety of therapeutic interventions [1,2,3]. Average patient survival even after the most advanced surgical treatments, or surgery combined with radiation or chemotherapy, remains dismal at only 12-18 months after initial diagnosis [3]. Cisplatin (CDDP), which intercalates into and cross-links DNA, has been widely used as a chemotherapeutic for GBM and other brain tumors [4,5]. Temozolimide (TMZ), a highly effective imidazotetrazine that was introduced more recently, alkylates DNA at guanine bases after hydrolysis, leading to single and double strand breaks [6,7]. Unfortunately, many tumors exhibit an inherent or acquired resistance to these chemotherapeutic agents, often necessitating doses that become cytotoxic to normal brain tissue [3]. Similar responses to radiotherapy may occur. Photodynamic therapy (PDT), which involves non-ionizing radiation, is less susceptible to pre-existing resistance and has emerged as one of the most attractive alternatives for treating brain malignancies [8,9,10,11]. An added advantage of PDT is that synergistic effects with chemotherapy (e.g., low-level cisplatin) are often possible, some of which are based on different subcellular sites of action. Whereas cisplatin and TMZ damage DNA in the nucleus, PDT typically causes cytoplasmic (mitochondrial, lysosomal, or ER) damage [12].

PDT was introduced about 45 years ago as a novel means of selectively eradicating a variety of solid tumors [13,14,15], many of which are refractory to conventional radiotherapy or chemotherapy. PDT is a minimally invasive modality, the classic version involving a pre-existing photosensitizing agent (PS), PS-exciting light in the visible-to-near infrared range, and molecular oxygen. All three of these factors must be engaged concurrently to produce cytotoxic reactive oxygen species (ROS). The most prominent of these ROS is singlet molecular oxygen (^1^O_2_), which can damage proteins, unsaturated lipids, and nucleic acids in target cancer cells [16]. A major advantage of PDT over chemotherapy or radiotherapy is that cytotoxic photodamage is typically limited to the tumor site itself and does not occur until PS, exciting light, and O_2_ are all engaged. Thus, light alone or PS alone is usually ineffective on a tumor, nor does it have any significant effect on normal tissue. Precise light delivery to a tumor via fiber optic transmitters provides an additional element of site-specificity during PDT [13,14,15]. Photofrin^®^, an oligomeric hematoporphyrin derivative, was the first administrable photosensitizer to receive FDA approval for PDT about 20 years ago and is now used for a variety of solid tumors, including brain tumors [14,15,17].

In more recently developed 5-aminolevulinic acid (ALA)-based PDT, ALA itself or an ALA ester is taken up by cancer cells and metabolized to protoporphyin IX (PpIX), the active PS, via the heme biosynthetic pathway [18,19]. In order to provide heme for rapid growth and progression, this pathway is more active in tumor cells than normal counterparts, although iron typically becomes limiting, allowing relatively large levels of PpIX to accumulate [19]. In cancer cells, PpIX accumulates initially in mitochondria, making these organelles primary targets of ALA-PDT damage that can kill cells via intrinsic apoptosis [18,19]. In addition to sensitizing cytotoxic reactions, ALA-induced PpIX produces a striking red fluorescence under relatively low-intensity exciting light. Many oncologists have exploited this aspect for image-guided resection (IGR), i.e., to clearly define the extremities of a tumor prior to its surgical removal, thereby greatly improving procedural accuracy [19]. Thus, ALA-induced PpIX, which is largely localized in tumor cells, has the advantage over most other anti-tumor agents of serving as a surgical guide on the one hand and a cytotoxic PDT sensitizer on the other.

## 2. Nitric Oxide and Its Multifaceted Roles in Cancer

Nitric oxide (NO) is a short-lived bioactive free radical (<2 sec in H_2_O) that diffuses freely on its own and, like O_2_, tends to partition into hydrophobic regions of cells, e.g., cell membranes [20]. NO is generated naturally by three different enzymes of the nitric oxide synthase (NOS) family: nNOS/NOS1 (neuronal), iNOS/NOS2 (inducible), and eNOS/NOS3 (endothelial). nNOS and eNOS usually function at relatively low constitutive levels, require Ca^2+^ for activation, and produce NO at nanomolar levels for short intervals. In contrast, iNOS can be induced by various stressors, does not require Ca^2+^ for activation, and can generate NO in the micromolar range for much longer periods [21,22]. All three NOS enzymes catalyze the conversion of L-arginine to citrulline and NO at the expense of NADPH and O_2_ [22]. It well established that eNOS-derived NO at low steady-state levels (1–10 nM) stimulates cyclic-GMP formation in vascular smooth muscle cells, leading to blood vessel relaxation. On the other hand, in activated macrophages during an immune response, iNOS-derived NO at much higher levels (1 µM or greater) is cytotoxic and potentially oncogenic [23,24,25]. Such effects typically occur after NO reacts with superoxide radical (O_2_^−^) to give peroxynitrite (ONOO^−^), a strong oxidant that can oxidatively damage DNA and membrane lipids. At intermediate levels (e.g., 50–300 nM), iNOS-derived NO can play a key role in cancer persistence and progression by activating oncogenic signaling pathways or inhibiting suppression pathways [26]. Thus, whether NO exhibits pro- or anti-tumor properties (Figure 1) depends largely on its steady-state levels, which are usually quite low (10–300 nM) in proliferating transformed cells. There is an increasing awareness that besides nNOS, most gliomas express iNOS and that iNOS-derived NO plays a major role in tumor cell survival, persistence, and progression [27,28,29]. For many malignancies, including gliomas, a direct correlation has been found between relatively high iNOS expression in tumor tissue and poor prognosis [29]. Proteins such as Survivin (which inhibits apoptosis [30]) and S100A4 (which stimulates invasion/metastasis [31]) can also serve as prognostic markers. It is important to note that in many cases, elevated expression of these markers might be a secondary effect of upregulated iNOS/NO, as demonstrated in a PDT challenge, for example (see below) [32].

As already pointed out, sub-micromolar nitric oxide (NO) in tumors plays a key role in tumor cell survival, persistence, and progression [23,24,25,26,27,28,29]. Such NO can also signal for resistance to radiotherapy, chemotherapy, or PDT [33,34,35,36,37]. The NO-mediated PDT resistance was first observed by Henderson et al. [36] and Korbelik et al. [37], using various syngeneic mouse tumors (breast, cutaneous, RIF carcinomas) sensitized with Photofrin^®^. It was found that tumor regression could be greatly improved by administering an iNOS activity inhibitor (e.g., L-NAME) immediately after PDT [37]. The extent of improvement correlated with constitutive NO production, tumors with a high output responding much better than those with a low output. The resistance was attributed mainly to NO-mediated relaxation of tumor blood vessels acting in opposition to PDT-induced vasoconstriction, i.e., a vascular effect of NO as opposed to a pro-survival effect on tumor cells per se [37]. A similar mechanism was deduced in subsequent studies by Reeves et al. [38], using ALA-based PDT on mouse tumor models. However, until relatively recently, many other questions pertaining to NO’s anti-PDT effects have remained unanswered, e.g., (i) the major cellular source of NO in any given tumor, e.g., cancer cells vs. stromal cells; (ii) the nitric oxide synthase (NOS) isoform that produces most of this NO, and (iii) the NO-mediated signaling events that lead to PDT resistance. Over the past eight years, these open questions have been addressed by the authors and colleagues using various cancer cell lines. Key findings from these studies will be discussed below, along with more recent findings dealing with NO’s negative effects on glioblastoma PDT.

## 3. iNOS/NO-Mediated Antagonism to PDT

As indicated in Section 1, one of the advantages of PDT is that it can often overcome any resistance that a tumor may have to chemotherapy or radiotherapy. Like these other modalities, however, PDT may elicit a resistance response (an acquired resistance) in many cancer cell types. One type of resistance, which involves iNOS-generated NO, was discovered in the authors’ laboratory. When subjected to an ALA-based photodynamic challenge in mitochondria, several human breast cancer lines (COH-BR1, MCF-7, MDA-MB-231) and prostate cancer lines (PC3, DU145) underwent intrinsic apoptosis that could be substantially enhanced by inhibitors of iNOS enzymatic activity (e.g., 1400W, GW274150) or by a selective NO scavenger (cPTIO) [39,40,41]. Seeing these enhancements implied that iNOS-derived NO was playing a key role in a hyper-resistance response. Moreover, this response was greatly attenuated when iNOS was depleted via shRNA-mediated knock down prior to ALA/light treatment [41]. More often than not, only iNOS (rather than other NOS isoforms) was upregulated by the challenge and this appeared to be more important in the hyper-resistance than any effect of the pre-existing enzyme [40,41,42,43,44]. As surviving cells continued to proliferate, a gradual decline in over-expressed iNOS was often observed, suggesting that elevated resistance was a transient phenomenon rather than selection for a relatively stable population of high iNOS expressing cells. Of added importance was our observation that breast and prostate cells surviving ALA/light treatment typically exhibited a more aggressive phenotype in terms of accelerated proliferation, migration, and invasion [43,44].

We recently extended the above in vitro findings to the in vivo level using immunodeficient female mice engrafted with breast MDA-MB-231 tumors [45]. After ALA administration, tumors were irradiated with red (633 nm) light, using an LED source. Treated animals exhibited a significant reduction in tumor growth compared with light-only controls over a two-day post-PDT period. However, 1400W in multiple post-PDT doses spaced one day apart reduced tumor growth much further, whereas it had no significant effect on control animals that were not ALA-treated before irradiation [45]. This suggested that pre-existing iNOS/NO had little, if any, effect on tumor resistance. Western blot analysis of tumor samples after PDT revealed a striking upregulation of iNOS and also a 1400W-inhibitable increase in the level of NO-derived nitrite [45]. This was the first reported evidence for iNOS upregulation by PDT in vivo using a human tumor model and for iNOS/NO-imposed resistance to tumor regression.

### 3.1. Post-PDT Upregulation of iNOS/NO in Glioma Cells: Increased Resistance to Photokilling

Like breast and prostate cancer cells [40,41,42,43,44], human glioblastoma U87MG cells (a.k.a. U87) that had been sensitized with ALA-induced PpIX underwent a progressive loss of MTT-assessed viability with increasing light fluence (Figure 2A). ALA alone or light alone without prior ALA treatment had no effect of viability. When sensitized cells were irradiated in the presence of 1400W or cPTIO, there was a striking increase in cytotoxicity with increasing light fluence, suggesting that endogenous iNOS/NO was signaling for resistance (Figure 2A). The extent of apoptosis after ALA/light treatment was also significantly greater when 1400W or L-NAME (a general NOS inhibitor) was present (Figure 2B), confirming iNOS/NO-elicited resistance [32]. As shown in Figure 2C(a), the level of iNOS protein in ALA/light (1 J/cm^2^)-challenged U87 cells increased progressively during post-irradiation incubation, reaching ~3-times the control level after 5 h. Not-surprisingly, nNOS was also expressed in U87 cells; however, it remained at the same starting level after cells were photodynamically stressed (Figure 2C(b)). This suggests that unlike iNOS, nNOS made no significant contribution to cytoprotection in these cells. Another established glioblastoma line (U251) responded similarly to ALA/light, exhibiting 4–5-fold iNOS overexpression over 20 h of post-irradiation incubation (Figure 2D), and a dramatic iNOS/NO-mediated hyper-resistance [32]. Thus, stress-induced iNOS/NO appeared to play a crucial role in the photostress-enhanced resistance of both glioblastoma cell types, U87 and U251.

A fluorogenic probe for NO (DAF-2DA) was used to establish whether photostress-induced iNOS and associated hyper-resistance was in fact due to iNOS-derived NO. After DAF-2DA’s cellular internalization and hydrolysis to DAF-2, the latter detects NO after its conversion to a nitrosating species such as N_2_O_3_ [46,47]. Within 4 h after an ALA/light challenge, U87 cells exhibited a strong NO-based fluorescence signal (>3-fold over an ALA-only background), which persisted for at least 20 h [32]. This signal was nearly abolished when 1400W or L-NAME was introduced immediately after irradiation, confirming that NO had been upregulated in photostressed U87 cells along with iNOS.

### 3.2. Accelerated Proliferation of PDT-Surviving Glioma Cells: Role of iNOS/NO

Cancer cells often adapt to stressful conditions by acquiring a more aggressive proliferative and migratory phenotype [1,2,3]. This proved to be the case for photodynamically stressed glioblastoma cells and iNOS/NO played a key driving role. As shown in Figure 3A, U87 cells that survived an ALA/light challenge 24 h after it was delivered exhibited a striking 2-fold growth spurt over the next 24 h relative to non-irradiated controls. This spurt was strongly attenuated by 1400W and also by cPTIO, clearly indicating iNOS/NO involvement, particularly that upregulated by the photostress (Figure 2C). The growth of non-irradiated control cells was slowed somewhat by 1400W (Figure 3A), suggesting a constitutive stimulatory effect of pre-existing iNOS/NO in U87 cells. This confirms the findings of others using non-stressed glioblastoma cells [48,49]. However, observing a robust iNOS/NO-dependent growth stimulation in response to therapy-based oxidative stress, in this case PDT [32], had not been described previously.

### 3.3. Role of iNOS/NO in Accelerated Migration of PDT-Surviving Glioma Cells

One other manifestation of hyper-aggressiveness in photostressed glioblastoma cells is more rapid migration into a cell-depleted space. Two other manifestations of hyper-aggressiveness were also observed in photostressed glioblastoma cells: (i) accelerated migration into a cell-depleted space, and (ii) accelerated invasion through an extracellular matrix (ECM)-like membrane. A gap-closure or “wound-healing” assay is typically used to examine forward migration into a voided zone generated by a straight-line scratch on a culture dish [50]. In a recent study, we photographed ALA-treated U87 cells in a scratch zone before irradiation and at various times after irradiation up to 24 h, during which cells were kept in the incubator. 1400W or cPTIO was included in the medium of certain dishes to test for iNOS/NO involvement in any altered migration. As shown by the gap-closure data in Figure 3B, ALA/light-stressed cells migrated more rapidly than ALA-only control cells over a 24 h post-irradiation period. This response was substantially blunted by 1400W, signifying major iNOS/NO dependency. 1400W also slowed control cell migration, but to a much smaller extent than in photodynamically-stressed cells, demonstrating the greater importance of stress-upregulated iNOS over basal iNOS in stimulating migration.

### 3.4. Role of iNOS/NO in Accelerated Invasion of PDT-Surviving Glioma Cells

A 96-place trans-well device was used to assess the invasiveness of U87 cells, i.e., ability to traverse a Matrigel-infused filter, moving from a serum-free upper well toward a serum-containing lower well [50]. Measurements commenced at 24 h after cell exposure to ALA alone (controls) or ALA plus irradiation. As shown in Figure 3C, 1400W inhibited the invasiveness of control cells by a small (barely significant) extent, suggesting iNOS/NO promotion of this basal activity. After irradiation, however, ALA-treated cells exhibited a striking 50% increase in invasion relative to control cells and 1400W abrogated this increase, demonstrating that iNOS/NO played a dominant role in the photostress-enhanced invasiveness.

Matrix metalloproteinases (MMPs) catalyze the degradation of ECM components such as collagen, laminin and fibronectin, and thus play a key role in cancer cell invasiveness and metastasis [51]. Type-9 MMP (MMP-9) is known to be actively involved in glioma cell migration/invasion [52] and this proved to be the case for U87 cells after an ALA/light challenge. Western blot analysis showed that there was no significant change in overall MMP-9 expression over a 24 h post-irradiation incubation period [32]. However, when in-gel zymography was used to monitor the activity status of externalized enzyme 24 h after irradiation, a dramatic 80% increase in activity was observed relative to a non-irradiated (ALA-only) control (Figure 3D). This activation was strongly depressed by L-NAME (>90%) and by 1400W (~70%), indicating that iNOS-derived NO played a prominent role in MMP-9 hyper-activation from its zymogen, pro-MMP-9 [32]. A fascinating sidelight to the MMP-9 response is that tissue inhibitor of metalloproteinase-1 (TIMP-1), which is known to be highly specific for MMP-9 [53], was progressively down-regulated in U87 cells after ALA/light treatment, as revealed by immunoblotting [32]. This response was strongly blunted by 1400W, pointing again to iNOS/NO dependency—in this case to prevent MMP-9 de-activation by TIMP-1. Of added interest is the fact that Survivin and S100A4, both known to play key roles in tumor cell growth, migration and invasion [30,31], were markedly upregulated in photostressed U87 cells, and in 1400W-inhibitable fashion [32]. Pro-metastatic S100A4 is particularly interesting because it was barely detectable in control cells, but reached a very high level 24 h after ALA/light, almost all of which was ablated by 1400W.

If occurring at the clinical PDT level, each of these negative responses to photostress-enhanced migration/invasion supported by MMP-9 activation and Survivin and S100A4 overexpression—would be problematic unless pharmacologically counteracted, e.g., by an inhibitor of iNOS expression or activity.

## 4. iNOS/NO-Induced Bystander Effects in PDT Models

Another dimension of post-PDT cancer cell aggressiveness was discovered recently: enhanced growth and migration of non-stressed bystander cells. In a tumor setting, bystander cells may lie in the general vicinity of cells targeted by a PDT or chemotherapeutic agent, but may not receive any (or enough) of it to elicit a therapeutic response, possibly due to insufficient vascular delivery. In the case of PDT, insufficient light delivery due to constraints of the light source and structural irregularities of the tumor can pose additional limitations. Most of the research relating to bystander effects has involved cancer-initiating vs. cancer-suppressing ionizing radiation (e.g., gamma rays, X-rays), which can produce effects ranging from DNA damage, mutations, and apoptosis to accelerated growth and migration of targeted cells [53]. Radiation-induced bystander effects can be transmitted via inter-cell gap junctions or via the medium, i.e., without physical contact between targeted and bystander cells [53]. Various signaling mediators have been proposed for ionizing radiation, including cytokines, H_2_O_2_, and NO, the latter receiving the greatest attention for bystander effects that occur independently of cell contact. NO produced specifically by radiation-targeted cells has been reported to elicit bystander effects ranging from DNA strand breaks and micronuclei formation to defective homologous recombination repair, leading to greater genetic instability, cell transformation, and accelerated proliferation [54]. Of special relevance here are early studies showing that X-ray-targeted glioblastoma cells overexpressed iNOS continuously over a 24 h post-radiation period, the resulting NO signaling for radioresistance in non-targeted bystander cells [55,56].

The possibility of bystander effects during non-ionizing PDT was first recognized about twenty years ago [57,58], but far less is known about this in mechanistic terms than its ionizing radiation counterpart. Hypothesizing that cells experiencing the greatest photodynamic (e.g., ALA/light) stress might send signals to non- or weakly-stressed bystanders, Bazak et al. [59,60] developed a novel approach for testing this. Two populations of sub-confluent cancer cells (initially prostate PC3 cells) were separated from one another on a large culture dish by two-to-four impermeable silicone-rimmed rings. The larger population (target cells, outside rings) was treated with ALA while the smaller population (bystander cells, inside rings) was not, after which the entire dish was irradiated. At some interval after irradiation, the rings were carefully removed, leaving a gap between the cell groups, and then both populations were analyzed for iNOS status, growth rate, and migration rate during dark incubation compared with non-irradiated (ALA-only) controls. Like surviving targeted cells, bystanders exhibited significant iNOS upregulation and accelerated growth and migration, each response being attributed to initial NO diffusing from the target compartment, based on the inhibitory effects of 1400W, cPTIO, or target cell iNOS knockdown [59]. Incubation of bystander cells with conditioned medium from ALA/light-targeted cells failed to induce the above responses, ruling out any involvement of relatively long-lived effectors such as cytokines, lipid peroxides, or NO-derived nitrite/nitrate. NO itself would not have survived in conditioned media because of its short lifetime (<2 sec) [20]. Thus, it had to be continuously generated by targeted cells to elicit bystander effects. Screening for possible effector proteins associated with enhanced PC3 bystander aggressiveness revealed a strong transient activation of the Akt and ERK1/2 kinases and an induction of cyclooxygenase-2 (COX-2), each response being cPTIO-inhibitable [59]. Although NO-mediated bystander effects for glioblastoma cells have not yet been assessed in a PDT format, they would most likely occur, based on evidence obtained with ionizing radiation [55,56]. PDT is becoming an increasingly attractive treatment option for glioblastoma and other brain malignancies [8,9,10,11]. Therefore, the negative implications of NO-mediated bystander effects (e.g., promotion of tumor growth and metastatic expansion) deserve serious attention aimed at mitigating these effects and improving treatment outcomes. One can view NO-mediated bystander effects as a type of relay or “feed-forward” phenomenon in which NO produced by targeted cells induces iNOS/NO in bystanders, and that this is propagated through the population. Thus, cells that escape being targeted in any given tumor are not necessarily unaffected by NO diffusing from neighboring targeted cells, a possibility that has not been well recognized up to now.

## 5. Mechanism of Glioblastoma iNOS Induction by Photostress: Preventative Approaches

In examining the underlying mechanism of iNOS upregulation in glioblastoma U87 cells after ALA/light exposure, Fahey et al. [61] found that transcription factor NF-κB played a crucial role. Accordingly, the active p65/RelA subunit of NF-κB translocated from cytosol to nucleus after photostress, and Bay11, an inhibitor of IKK which phosphorylates and releases restraining subunit IκB on NF-κB, prevented p65 translocation. Importantly, Bay11 nearly abolished all iNOS expression (basal as well as photostress-induced), clearly linking this with p65-mediated iNOS gene transcription [61]. Similar findings were reported earlier for a breast cancer cell line [42]. Based on the non-glioblastoma work of others [62,63], we postulated that acetylation (ac) of specific lysine (K) residue(s) on p65 played a key driving role in iNOS transcription. Such acK modifications can be recognized by one or more proteins of the bromodomain and extra-terminal domain (BET) family, also known as epigenetic “readers” [64,65]. BET proteins are known to interact with histones and transcription factors at acK sites, thereby co-activating transcriptional processes [64,65,66]. In testing our hypothesis using pull-down assays, we found that BET-4 protein (Brd4), but not its BET-2 (Brd2) analogue, interacted strongly with NF-κB p65 in U87 cells that had been ALA/light-stressed [61]. Of added importance were our findings that the levels of Brd4 and acK310 on p65 each increased ~3-fold in photostressed cells, whereas the Brd2 and p65 levels were unaffected. Thus, stress-elevation of Brd4 and p65-ack310 would have promoted interaction of these proteins for augmented translational activity at the iNOS promoter.

Several BET protein inhibitors have recently emerged as highly potent and specific pharmacologic suppressors of cancer cell proliferation and invasive/metastatic expansion [67,68]. These inhibitors function by binding to acK recognition domains on BET proteins and preventing their interaction with histones or transcription factors. Several of these agents have advanced to clinical trials for various malignancies, including lymphomas, myelomas and triple negative breast cancers [67]. The thienodiazepine JQ1 is a prototypical BET inhibitor that has repressed cancer cell progression in many in vitro and in vivo models, including glioblastoma models [69]. It was recently reported that JQ1 blocks the TNF-α-stimulated expression of several pro-survival genes in lung cancer cells by binding and inactivating Brd4 [63]. However, iNOS was not included among the gene products described, prompting us to investigate this in our PDT studies on glioblastoma cells. At a very low concentration (0.3 µM), JQ1 by itself had little effect on U87 cell viability, but it enhanced ALA/light-induced apoptosis in synergistic fashion [61]. At the same time, JQ1 nearly abolished Brd4-p65 interaction after ALA/light treatment along with total iNOS expression - basal as well as stress-upregulated. However, it had no effect on the extent of Brd4 or p65-acK310 elevation, implying that these responses were not under JQ1-iinhibitable transcriptional control. A striking loss of photostress-enhanced U87 cell aggressiveness was also observed when JQ1 was present. For example, JQ1 slowed surviving cell proliferation rate by >80% compared with ~50% for 1400W, but more impressively, it did so at only ~1% the 1400W concentration in bulk cell suspension [61]. The BET inhibitor also suppressed the greater invasiveness of glioblastoma cells that survived an ALA/light challenge. Results for U87 and U251 cells were similar; the latter is represented in Figure 4. Active JQ1 reduced invasion rate of vehicle control (DMSO) by ~10% [61], whereas the inactive enantiomer [JQ1(-)] had no effect. PDT-surviving cells invaded ~40% faster than controls (Figure 4) and JQ1 (but not JQ1(-)) not only abolished this acceleration but reduced the rate to ~35% that of the control. Bay 11 at ~20-fold higher concentration than JQ1 also had a large inhibitory effect, but specificity could be an issue because Bay11 can inhibit other pro-survival effectors besides Iκκ [70]. Like JQ1, 1400W also eliminated stress-enhanced invasion but left residual activity closer to the control value (Figure 4). Consequently, the effects of an iNOS transcriptional inhibitor (JQ1) were clearly much more impressive than those of an iNOS activity inhibitor (1400W), the former operating at ~100-times lower concentration than the latter. Of added interest is the observation that photostress-upregulation of other NF-κB-regulated proteins besides iNOS (e.g., Survivin, Bcl-xL, MMP-9) was also blocked by JQ1, but whether this occurred directly or secondarily in some cases as a result of iNOS/NO suppression [61] is not yet clear, at least for glioblastoma cells. A scheme depicting Brd4-p65_ac_-dependent transcriptional upregulation of iNOS/NO by ALA-PpIX/light-induced stress and its suppression by JQ1 is shown in Figure 5.

More recent studies have focused on events upstream of NF-κB activation and iNOS/NO upregulation in photodynamically-targeted glioblastoma cells. Some key findings were that (i) ^1^O_2_ generated by photoactivation of ALA-induced PpIX in mitochondria was indirectly involved in activation of the cytosolic pro-survival PI3K/Akt kinase system in U87 cells; (ii) Tumor suppressor PTEN was rapidly inactivated by intramolecular disulfide bond formation during ALA/light treatment, thereby fostering PI3K activation; (iii) transacetylase p300 was phosphorylation-activated by Akt, leading to greater acK310 formation on NF-κB/p65; (iv) the latter was facilitated by greater p300-p65 interaction as demonstrated by pull-down assay; (v) the deacetylase Sirt1 was down-regulated in ALA/light-treated U87 cells, consistent with elevated p65-acK310 levels; and (vi) chemical inhibitors of upstream PI3K and p300 activity also inhibited iNOS upregulation under photodynamic stress [71].

## 6. Anti-Cancer Potential of NO at Relatively High Levels

Recent studies by Rapozzi et al. [72,73], using melanoma cells photosensitized with pheophorbide-a, have supported our findings regarding the cytoprotective effects of iNOS/NO on ALA/light-stressed glioma cells. Thus, low-level NO from low PDT light doses was found to be cytoprotective through anti-apoptotic activation of the NF-κB/Snail/RKIP pathway [72]. In contrast, higher NO levels from relatively high light doses proved to be cytotoxic due to pro-apoptotic activation of this pathway [72,73]. Whether a significant elevation of iNOS/NO via greater PDT pressure might produce a similar NO-enhanced proapoptotic response in glioblastoma cells has not yet been determined.

The anti-tumor potential of exogenous high-level NO, either alone or in combination with other treatments, is being explored by various investigators. NO donors such as glyceryl trinitrate, sodium nitroprusside, S-nitrosoglutathione (GSNO), diazeniumdiolates (NONOates), and Lopinavir-NO have been tested in this way with varying degrees of success, a key question being whether undesirable (off-target) effects occur [74,75,76]. The latter question has been addressed by the development of donors such as JS-K, which releases NO upon activation by glutathione S-transferases [77]. These enzymes are expressed at relatively high levels in many malignant cells, thus increasing the specificity of NO delivery. Cytotoxic NO donors have not yet been tested on gliomas. However, based on our own studies [32,39,40,41,42,43,44,45], some concerns arise, e.g., that any long-lasting release of NO in relatively low fluxes might actually enhance tumor aggressiveness rather than suppress it. Such concerns could be mitigated with the development of rapid NO release photosensitizing agents [78,79]. For example, the nitrosyl phthalocyanine ruthenium complex (Ru-NO) generates both ^1^O_2_ and NO upon photoexcitation [78]. This complex was found to be more cytotoxic than a ^1^O_2_-only control, suggesting that rapid photorelease of NO in high fluxes with little or no persistent low-flux release could provide major advantages for PDT [78,79]. Production of high dose NO by light-activated donors has also been shown to promote the chemotherapeutic effects of doxorubicin by inhibiting its export via ATP binding cassette (ABC) transporters [80]. A similar approach might benefit ALA-PDT for brain tumors, given the recent evidence that the ABCG2 transporter promotes efflux of ALA-induced PpIX from brain cancer cells [81].

## 7. Summary and Conclusions

Like two other gasotransmitters, carbon monoxide (CO) and hydrogen sulfide (H_2_S), NO is known to have normo-physiologic as well as patho-physiologic roles, depending in large part on its generation rate and local concentration [82]. Many malignant tumors, including highly aggressive and lethal glioblastomas, exploit low flux NO to promote survival, stimulate proliferation and migration/invasion, and to resist eradication by PDT and other therapeutic interventions. PDT is now considered one of the most promising of the new anti-glioma therapies [8,9,10,11]. ALA-based PDT is attracting particular attention because tumor cells are hyperactive in heme biosynthesis and generate higher levels of ALA-derived PpIX than normal cells, thereby increasing tumor vulnerability to photodynamic action. Of added importance, PpIX fluorescence at low light intensities can be exploited for image-guided resection, i.e., to better define tumor boundaries during surgery [83]. In this review, we have discussed various aspects of iNOS/NO antagonism to ALA-PDT ranging from tumor cell resistance to photokilling to accelerated proliferation, migration and invasion of cells that withstand the challenge. Although these negative effects can be observed in directly targeted cells, they may also develop in non-targeted bystander cells as NO from the former diffuses to the latter and iNOS/NO is upregulated there as well [59]. Although we have emphasized glioblastoma PDT here, various human breast and prostate cancers respond similarly to this type of challenge, employing endogenous iNOS as the major source of signaling NO [84]. Based on substantial evidence, this NO typically derives from stress-upregulated iNOS rather than the pre-existing or basal enzyme. Other oxidative therapeutics may also give rise to iNOS/NO-dependent hyper-resistance and/or hyper-aggressive phenotypes, one example being cis-platin-treated prostate PC3 cells, which also exhibit bystander effects (Fahey and Girotti, unpublished observations).

How endogenous NO exerts anti-PDT effects is still not clear in chemical mechanistic terms. S-nitrosation of select cysteine residues on key effector proteins might initiate resistance signaling [23,24,25,26,85], but this modification can be transient and difficult to define in PDT systems such as we describe. Thus, much remains to be learned in terms of underlying chemical biology. Another key issue relates to how iNOS/NO’s anti-therapeutic effects might be overcome by pharmacologic intervention. As discussed in the context of glioblastoma PDT, the BET inhibitor JQ1 suppressed iNOS transcription (Figure 5) and reduced acquired hyper-aggressiveness much more powerfully than an inhibitor of iNOS activity. Thus, JQ1, which has tested positively against glioblastoma on its own [61], might significantly improve therapeutic outcomes for this malignancy when combined with PDT or possibly some other modality that stimulates iNOS/NO. Transcriptional upregulation of pro-survival/expansion iNOS under therapeutic stress like PDT may occur more often than currently recognized, thus emphasizing the need for powerful inhibitors like JQ1 as therapeutic adjuvants.

## Figures and Tables

**Figure 1 cancers-11-00231-f001:**
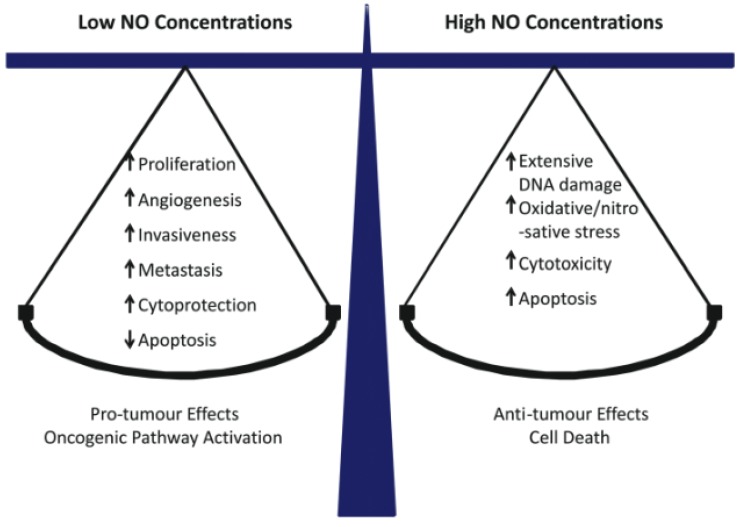
Concentration-dependent effects of NO in cancer: pro-tumor versus anti-tumor. Reproduced from Reference [24], with permission.

**Figure 2 cancers-11-00231-f002:**
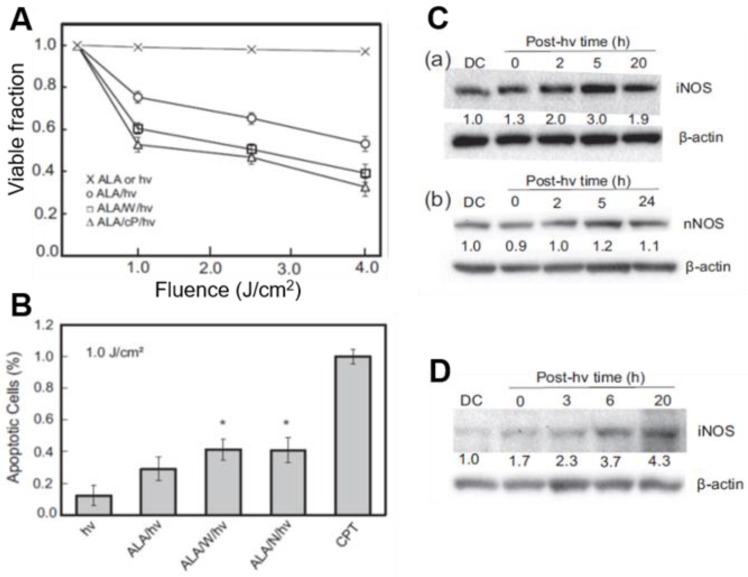
Upregulation of iNOS/NO in photodynamically-stressed glioblastoma cells confers resistance to photokilling. U87 or U251 cells at ~60% confluency were dark-incubated with 1 mM ALA for 30 min in serum-free medium, irradiated with broad-band visible light, then switched to serum-containing medium and dark-incubated for various periods. Where indicated, 25 µM 1400W (W) or cPTIO (cP) was introduced immediately after irradiation and kept at the same concentration throughout. (**A**) MTT-assessed U87 viability 20 h after ALA/hν exposure in absence vs. presence of W or cP; *n* = 3. (**B**) Annexin V-FITC-assessed U87 apoptosis 4 h after indicated treatments (CPT: camptothecin); *n* = 3, * *p* < 0.05 vs. ALA/hν. (**C**) Immunoblot of U87 iNOS (a) and nNOS (b) at indicated post-hν times. Numbers below NOS bands indicate intensity relative to actin and normalized to dark (ALA-alone) control (DC). (**D**) Immunoblot of iNOS in U251 cells at indicated post-hν times. Adapted from Reference [32].

**Figure 3 cancers-11-00231-f003:**
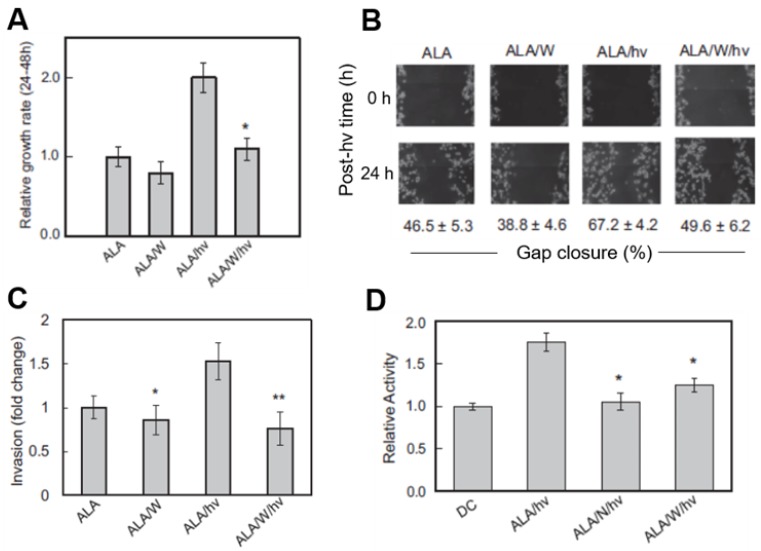
Enhanced aggressiveness of glioblastoma cells that survive a photodynamic challenge. U87 cells at ~40% confluence were sensitized with ALA-induced PpIX and irradiated (ALA/hν: light dose/fluence ~1 J/cm^2^). Where indicated, 25 µM 1400W (W) or 1 mM L-NAME (N) was introduced immediately after irradiation and maintained as such thereafter. Dark controls (ALA or ALA/W) were run alongside. After 24 h of post-hν incubation, any detached/dead cells were washed off and aggressive properties of remaining live cells were determined. (**A**) MTT-assessed proliferation rate; means ± SEM, *n* = 3, * *p* < 0.01 vs. ALA/hν. (**B**) Gap-closure-assessed migration rate; means ± SEM, *n* = 3.(**C**) Trans well chamber-assessed invasion rate; means ± SEM, *n* = 4, * *p* < 0.05 vs. ALA; ** *p* < 0.0001 vs. ALA/hν. (**D**) Gel zymography-assessed MMP-9 activity; means ± SEM, *n* = 3, * *p* < 0.01 vs. ALA/hν. Adapted from Reference [32].

**Figure 4 cancers-11-00231-f004:**
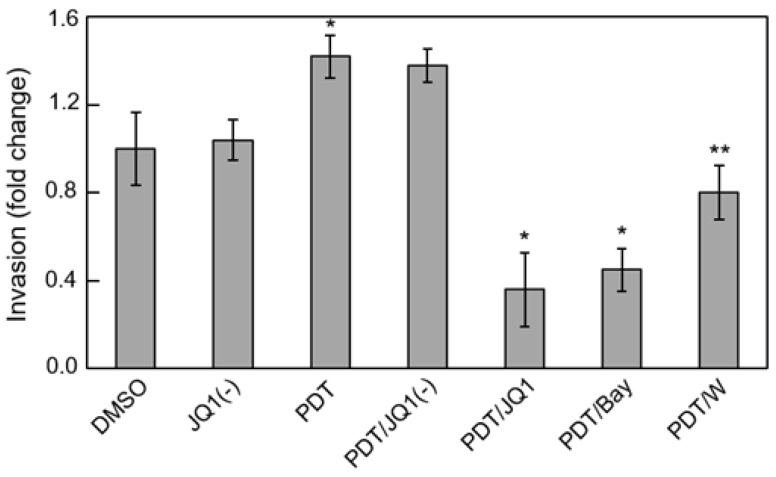
JQ1 abrogation of photostress-enhanced glioblastoma cell invasiveness. U251 cells at ~60% confluency were pre-incubated with 1 mM ALA. After PDT irradiation (1 J/cm^2^), these cells, along with dark controls, were harvested and assessed for invasiveness in the absence vs. presence of 0.3 µM JQ1, 0.3 µM JQ1(−), 5 µM Bay11, or 25 µM 1400W. DMSO served as a vehicle control for JQ1 and JQ1(−). Cells crossing Matrigel-infused filters after an incubation period of 24 h were pelleted and quantified by CCK-8 assay. Plotted invasion values are means ± SEM (*n* = 3); * *p* < 0.001 vs. PDT; ** *p* < 0.01 vs. PDT. Reproduced from Reference [61], with permission.

**Figure 5 cancers-11-00231-f005:**
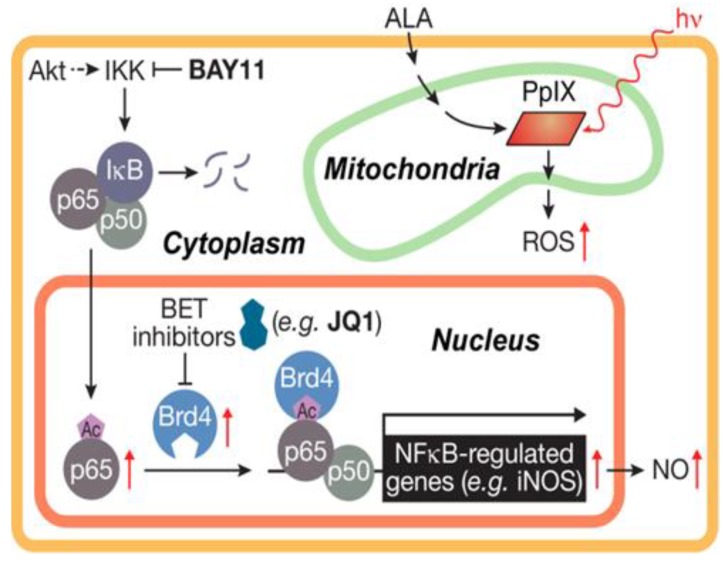
Scheme showing (i) photosensitized generation of ROS (^1^O_2_) by ALA-induced PpIX, (ii) activation of NF-κB, (iii) nuclear translocation and acetylation of p65, and (iv) JQ1-inhibitable transcription of iNOS as mediated by co-activator Brd4. Reproduced from Reference [61], with permission.

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
