# Peer review of "Nitric Oxide Antagonism to Anti-Glioblastoma Photodynamic Therapy: Mitigation by Inhibitors of Nitric Oxide Generation"

_cancers, 2019, doi:10.3390/cancers11020231_

Round 1
Reviewer 1 Report
Dear authors
This review paper is mentioned so much interesting topic and well organized with good contents to understand a relationship between nitric oxide (NO) and anti-glioblastoma photodynamic therapy (PDT). Therefore, it is suitable for publication in this journal after answer for following questions and minor revision as below comments.
Question 1. Page 6, in Figure 3, the authors used low light dose/fluence (~1 J/cm2). What about higher light dose/fluence of 2.5 and/or 4 J/cm2 like in Figure 2A??? Are there any different result? Also is there any special reason to use the light dose/fluence???
Question 2. PDT result is so much sensitive with NO concentration, however, it is not easy to control NO concentration in tumor site. Could you suggest any good way/method to get easy control of NO concentration in tumor environment???
Question 3. NO effect in cancer progression result is significantly depends on NO concentration as well as tumor cell type. In this paper the authors focused on NO concentration using glioblastoma cells. What about other cell type??? How do you expect?
Comments:
1. Page 1, lines 33-38 for PDT, move to line 52 in page 2.
2. Page 2, before line 52 "In more recently developed...", make a new paragraph for ALA-PDT.
3. Figures 1, 2, and 3 have low quality images, therefore it is need to make high quality images.
4. Page 3 in Figure 1 caption, add all sentences of the figure caption from Ref. 24 to understand concentration levels.
5. Pages 4 and 5 in Figure 2, change Figure 2C(a) to Figure 2C, Figure 2C(b) to Figure 2D, Figure 2D to Figure 2E to make clear images. Therefore, change related sentences in page 4 and figure caption in Figure 2.
6. Page 5, lines 172~178, for fluorogenic probe, I suggest that add one figure (as Figure 3) for easy understanding. Therefore, change Figure 3 in page 6 to Figure 4.
7. This review paper is focused on anti-glioblastoma, therefore, it is need to add "glioma" in the titles of sections 3.2, 3.3, and 3.4. For example, 3.2. ...of PDT-surviving glioma cells: role of iNOS/NO.
8. Page 7, Section 4 is not clear that ... bystander effects ... is for all cancer type? (Refs. 59, 60) or only for glioblastoma? (Refs. 55, 56) Also there is no figure to explain the Section 4, therefore it is need to add one figure (as Figure 5) as well as to add/correct related sentences.
9. Page 8, for the first paragraph of Section 5, also need to add related figure for understanding mechanism. I suggest that modify Figure 5 in page 10 (as Figure 6), and then move to page 8 after the first paragraph. Therefore, it is need to reorder related sentences and to correct Figure 4 in page 9 to Figure 7.
10. Page 10, Figure 5 should be corrected to show JQ1 effect (BET inhibitors), resulting in decrease of iNOS (iNOS ↓) and NO (NO ↓). Also in Figure 5, it is need to add (i), (ii), (iii), and (iv) in suitable places.
11. Page 10, lines 374 and 393, [Fahey and Girotti, unpublished results/observations] are not suitable as citations.
12. Please make sure there are no error letters and wrong space, etc.
Thanks so much.
Author Response
We thank the reviewer for the helpful suggestions and respond as follows:
Question 1. As shown, most experiments testing effects of PDT on iNOS level, proliferation, and migration/invasion of surviving glioblastoma U87 cells 20 h after irradiation were done at 1 J/cm2 fluence. However, a previous study with breast cancer cells (Ref. 40) showed that all these responses were greater when 2 J/cm2 was used compared with 1 J/cm2 - as might be expected, but naturally the surviving fraction was lower at 2 J/cm2. Similar results would be expected with U87 cells.
Question 2. As we indicated, the best way to lower PDT-induced NO in tumors is by using an INOS enzyme inhibitor (e.g. 1400W), an iNOS transcription inhibitor (e.g. JQ1), or an NO trap (cPTIO). As far as we know, high specificity via direct tumor targeting of these agents is not available now, but PDT-stressed tumor makes more iNOS/NO than normal tissue, so this would favor NO control by these agents.
Question 3. Regarding other cancer cell types, our earlier work (Refs. 39-44) with human breast and prostate cancer lines gave responses (PDT-stimulated iNOS, growth, and migration/invasion) like those for glioblastoma cells. So these appear to be common negative responses to anti-tumor PDT.
Comments
1. Lines 32-38 are on second page of Introduction and cannot be moved as suggested in order to retain their meaning.
2. OK, we made a new paragraph for ALA-PDT.
3. Unfortunately, we no longer have original images for Figs. 1-3. However, the ones provided seem clear enough for accurate reading.
4. We're sorry, but this is unnecessary because relevant concentrations are given here and in Fig. 1.
5. We don't agree. The given arrangement is clear and best for Fig. 2C, i.e. 2C(a) and 2C(b).
6. The reviewer makes a good point about the NO probe, so we have now included a reference [32] to our use of this probe, DAF-2DA. We feel that this reference [32] is enough: a new figure isn't necessary.
7. OK, we agree, and have inserted "glioma" to indicate cell type.
8. In Sect. 4, we indicate that all of our published bystander work has been done on prostate cancer PC3 cells, but we indicate that glioma cells should behave similarly under PDT. We don't think it's appropriate to include a figure since our data thus far are only on prostate cells.
9. We're sorry, but we do not fully understand these suggestions or why they are really necessary. The mechanism of iNOS induction and inhibition of iNOS transcription is well illustrated in Fig. 5.
10. Sorry, but JQ1 effects on iNOS/NO are in fact well illustrated in Fig. 5.
11. OK, these results are now indicated as "in press" (Ref. 71).
12. OK, lettering and spacing have been checked and are correct throughout to best of our knowledge.
Reviewer 2 Report
This is a very comprehensive review prepared from Authors with direct and proven experience in this area of research. The possible use of specific NO inhibitors during anti-glioblastoma photodynamic therapy and possibly during photodynamic therapy for other cancers is of great interest.
I have only a few comments on this paper as it stands.
1. When the Authors discuss the pro vs anti oncogenic properties of endogenous NO, I would advise them to emphasize more that the anti-oncogenic role of NO has attracted sufficient interest to generate NO-releasing drugs that have shown promising anti-cancer effects in in vitro and in vivo settings, including glioblastoma models
Basile MS et al., Anticancer and Differentiation Properties of the Nitric Oxide Derivative of Lopinavir in Human Glioblastoma Cells.Molecules. 2018 Sep 26;23(10). pii: E2463. doi
Maksimovic-Ivanic D et al., Int J Cancer. 2017 Apr 15;140(8):1713-1726. doi: 10.1002/ijc.30529. Epub 2017 Jan 20.
Maksimovic-Ivanic D et al.,Anticancer properties of the novel nitric oxide-donating compound (S,R)-3-phenyl-4,5-dihydro-5-isoxazole acetic acid-nitric oxide in vitro and in vivo.Mol Cancer Ther. 2008 Mar;7(3):510-20. doi: 10.1158/1535-7163.MCT-07-2037.
Huang Z et al.,JS-K as a nitric oxide donor induces apoptosis via the ROS/Ca2+/caspase-mediated mitochondrial pathway in HepG2 cells.Biomed Pharmacother. 2018 Nov;107:1385-1392. doi: 10.1016/j.biopha.2018.08.142. Epub 2018 Aug 31.
Huang Z et al.,Nitric Oxide Donor-Based Cancer Therapy: Advances and Prospects.J Med Chem. 2017 Sep 28;60(18):7617-7635. doi: 10.1021/acs.jmedchem.6b01672. Epub 2017 May 23.
Seabra AB, Durán N. Nitric oxide donors for prostate and bladder cancers: Current state and challenges.Eur J Pharmacol. 2018 May 5;826:158-168. doi: 10.1016/j.ejphar.2018.02.040. Epub 2018 Mar 1.
2. Among beneficial effects of NO in cancer , the Authors should indicate the inhibition by NO and NO releasing compounds of ATP-binding cassette (ABC) transporters P-glycoprotein (P-gp), multidrug resistance-associated protein 1 (MRP1), and breast cancer resistance protein 1 (BCRP1). This action of exogenous or endogenous NO may prevent chemotherapeutic resistance and the Authors should discuss briefly what they think the unwanted effects (in this or other chemotherapeutic pathways induced by NO) of specific inhibition of NO during PDT coud be in cancer patients.
Rothweiler F et al.,Anticancer effects of the nitric oxide-modified saquinavir derivative saquinavir-NO against multidrug-resistant cancer cells.Neoplasia. 2010 Dec;12(12):1023-30.
Chegaev K et al., Light-Regulated NO Release as a Novel Strategy To Overcome Doxorubicin Multidrug Resistance.ACS Med Chem Lett. 2017 Jan 30;8(3):361-365. doi: 10.1021/acsmedchemlett.7b00016. eCollection 2017 Mar 9.
3. The Authors should acknowledge and discuss that due to chemotherapeutic and anti-oncogenic properties of NO , also opposite approaches to what they suggest have been successfully investigated that entail Combination of PDT photosensitizers with NO photodononors
Fraix A, Sortino S.Combination of PDT photosensitizers with NO photodononors.Photochem Photobiol Sci. 2018 Nov 1;17(11):1709-1727. doi: 10.1039/c8pp00272j. Epub 2018 Aug 24.
4. The Authors should quote and discuss a recent review on the role of endogenous gases related to NO such as H2S and CO in biological processes. I think it could be interesting to mention that NO is not the only endogenous gas to regulate oncogenesis and that often the buiological actions of NO are related and coordinated with other endogenous gases such as H2S and CO. Studying the role H2S and CO during PDT antiglioblastoma therapy may add further therapeutic insights.
Fagone P et al.,Gasotransmitters and the immune system: Mode of action and novel therapeutic targets.Eur J Pharmacol. 2018 Sep 5;834:92-102. doi: 10.1016/j.ejphar.2018.07.026. Epub 2018 Jul 20.
Author Response
We thank Reviewer 2 for the valuable suggestions and comments, and respond to them as follows:
Comment 1. Pro- versus anti-oncogenic properties of exogenous NO: Yes, we agree that inclusion of key references about anti-tumor (high level) NO from non-light or light-driven donors is important as a counterpart to our main discussion about endogenous low level (pro-tumor) NO. Accordingly, we have written a new section (Sect. 6, p. 15) which incorporates and discusses some of the suggested references: new Refs. 75 and 76. We have also included some other references dealing with exogenous anti-cancer NO: Refs. 72-74,77, 78.
Comment 2. Inhibition of ATP-binding cassette (ABC) transporters by NO and NO-releasing compounds: Suggested work by Chegaev et al. is described now in Sect. 6 (new Ref. 80), along with a paper indicating that export of ALA-induced PpIX is mediated by ABCG2 (Ref. 81) in glioma cells.
Comment 3. Combination PDT and NO photodonors: The Fraix/Sortino paper is now mentioned in Sect. 6 (Ref. 79).
Comment 4. NO as a gasotransmitter along with H2S and CO: The paper by Fagone et al. is now mentioned in the Summary section, p. 19, Ref. 82.
Round 2
Reviewer 2 Report
The Authors have appropriately addressed my criticisms